



# Measurement report: Nocturnal subsidence behind the cold front enhances surface particulate matter in the plain regions: observation from the mobile multi-lidar system

Yiming Wang[1, 2], Haolin Wang[1, 2], Yujie Qin[1, 2], Xinqi Xu[1, 2], Guowen He[1, 2], Nanxi Liu[1, 2], Shengjie Miao[1, 2], Xiao Lu[1, 2], Haichao Wang[1, 2, *], Shaojia Fan[1, 2, *]

[1]School of Atmospheric Sciences, Sun Yat-sen University, and Southern Marine Science and Engineering Guangdong Laboratory (Zhuhai), Zhuhai, 519082, China
[2]Guangdong Provincial Observation and Research Station for Climate Environment and Air Quality Change in the Pearl River Estuary, Key Laboratory of Tropical Atmosphere-Ocean System (Sun Yat-sen University), Ministry of Education, Zhuhai, 519082, China

*Correspondence to:*
*Haichao Wang (wanghch27@mail.sysu.edu.cn); Shaojia Fan (eesfsj@mail.sysu.edu.cn)*

**Abstract.** A multi-lidar system, mounted in vehicle to monitor the profiles of temperature, wind and particle optical properties, was utilized to investigate the winter fine particulate matter ($PM_{2.5}$) pollution for a vertical perspective, in four cities in China in winter 2018. We observed the enhancement of surface nocturnal $PM_{2.5}$ in two typical plain cities (Changzhou and Wangdu), which was attributed to the subsidence of $PM_{2.5}$ transported from upstream polluted areas, with the wind turning north and downdrafts dominating. Combining with the observed surface $PM_{2.5}$, the reanalysis meteorological data, and the GEOS-Chem model simulation, we revealed the Transport-Nocturnal $PM_{2.5}$ Enhancement by Subsidence (T-NPES) events occurred frequently in the two cities, with percentages of 12.2 % and 18.0 %, respectively during Dec. 2018 - Feb. 2019. Furthermore, the GEOS-Chem model simulation further confirmed that the ubiquity of winter T-NPES events in a large scale including North China Plain and Yangtze River Delta. Process analysis revealed that the subsidence was closely correlated with the southeasterly movement of the high-pressure system and the passage of the cold front, resulting in the increase of temperature aloft, a stronger inversion layer, and further $PM_{2.5}$ accumulation in the atmospheric boundary layer. Thus, a conceptual model of the T-NPES events was proposed to highlight this surface $PM_{2.5}$ enhancement mechanism in these plain regions. However, it was not applicable to the two cities in basin region (Xi'an and Chengdu), due to the obstruction of the weather system movement by the mountains surrounding the basin.



## 1 Introduction

The severe fine particulate matter ($PM_{2.5}$, particles with an aerodynamic diameter smaller than 2.5 μm) pollution, caused by the rapid industrialization and urbanization in China (Guo et al., 2014; Huang et al., 2014), has essential impacts on visibility, ecosystem, regional and global climates, and human health (Yue et al., 2017; An et al., 2019; De Marco et al., 2019; Li et al., 2019b; Hao et al., 2021). To mitigate the $PM_{2.5}$ pollution, the government of China has implemented the Air Pollution Prevention and Control Action Plan in 2013 by strict emission controls (Gao et al., 2020). Despite the annual average concentration of $PM_{2.5}$ has been significantly decreased (Ding et al., 2019; Li et al., 2019a; Zhang et al., 2019b; Silver et al., 2020; Geng et al., 2021b), the $PM_{2.5}$ levels in the majority of Chinese cities are still above the World Health Organization target (WHO. 2021). Particularly, the issue of $PM_{2.5}$ pollution remained critical in the North China Plain (NCP) and Yangtze River Delta (YRD) in winter time (Peng et al., 2021; Qin et al., 2021).

The formation mechanisms of $PM_{2.5}$ pollution were complex especially in China (Guo et al., 2014; Xiao et al., 2021b). Such as the high emission intensity (Zhang et al., 2019b), the rapid chemical formation of secondary particles owing to the gas-phase and heterogeneous reactions (Wang et al., 2017; Lu et al., 2019; Chen et al., 2020), and the interactions within the atmospheric boundary layer (ABL) (Ding et al., 2013; Gao et al., 2016; Dong et al., 2017; Li et al., 2017). While the long-range transport also had significant impacts on the $PM_{2.5}$ pollution (Guo et al., 2014; Zhang et al., 2015; Huang et al., 2018). Cold fronts, as a common synoptic circulation in winter, were usually favourable for the quick removal of the locally accumulated pollutants in the NCP (Zhao et al., 2013; Gao et al., 2017), but conversely transport the pollutants to the YRD through a long distance (Kang et al., 2019; Huang et al., 2020; Kang et al., 2021). Zhou et al. (2023) indicated that the cold fronts could transport the precursors to the residual layer, where the secondary pollution was rapidly driven to be generated and then exacerbate near-surface air pollution as a result of the development of the daytime convective ABL. However, the above studies have focused on the impact of the horizontally transported pollutants on the downstream regions after the passage of the cold front. While few studies have been conducted on the variation in the vertical direction of particulate matter in the ABL during the passage of the cold front.

The vertical mixing exchange process between layer has great impacts on local air quality and the subsidence motion is associated with the evolution of inversion layer (Gramsch et al., 2014; Xu et al., 2018; He et al., 2022). Zhang et al. (2022) reported that the $PM_{2.5}$ concentration behind the cold front increased due to the subsidence motion and inversion layer. Zhao et al. (2023) suggested that the frontal



downdrafts were an additional transport pathway in the nighttime to make higher contribution to the
ground nitrate. Both of their studies were based on the model simulations, the observational evidence of
the subsidence behind the cold front and its impact on the nocturnal $PM_{2.5}$ enhancement events is still
lacking. Shi et al. (2022) reported one subsidence case of particulate matter during the passage of the
cold front over Wangdu, China in winter, which revealed that the subsidence was closely connect to the
enhancement of nocturnal $PM_{2.5}$.
To investigate the mechanisms of nocturnal $PM_{2.5}$ enhancement triggered by subsidence, the three-
dimensional spatial and temporal distribution is crucial. Many field observations of the vertical
distribution of particulate matter have been performed employing various methods such as tethered
balloons (Wang et al., 2021; Ran et al., 2022), airplane (Wang et al., 2018; Fast et al., 2022), unmanned
aerial vehicles (Song et al., 2021; Dubey et al., 2022) and the meteorological towers (Li et al., 2022; Yin
et al., 2023). Lidar, as an active remote sensing device with high temporal and spatial resolution, has
been extensively employed in atmospheric detection to obtain the profile of particulate matter, wind, and
temperature. The ground-based and satellite-based lidar have been widely used to detect the vertical
distribution of aerosol. In recent years, the mobile multi-lidar system has been gradually developed and
has become a powerful tool to observe the development of target species detection in a vertical
perspective. Compared with the traditional ground-lidar system, the mobile multi-lidar system enables
continuous mobile observations and provides information on the distribution of specific factors along its
path and can be used as an effective supplement to other fixed lidars. Additionally, the mobile multi-
lidar system can reach different cities by its portable setting in a short time to carry out the fixed-point
observations. The mobile lidar system had been used to carry out several observations in the past few
years (Lv et al., 2020; He et al., 2021; Xu et al., 2022). He et al. (2021) investigated the vertical
distribution characteristics of particulate matter in the Guanzhong Plain by using the mobile multi-lidar
system. Xu et al. (2022) conducted an observational study on the three-dimensional structure of
particulate matter distribution in the Guangdong-Hong Kong-Macao Greater Bay Area by using the
mobile multi-lidar system and proposed a conceptual model to elucidate the vertical distribution of
particulate matter under different wind and temperature conditions.
Here, we conducted the first nationwide field measurements in winter 2018 using the mobile multi-lidar
system during winter 2018 in China, to investigate the vertical distribution characteristics of particulate
matter in different cities. We focus on the observed nocturnal $PM_{2.5}$ enhancement events and seek insights
into their characteristics and the causes, by combining the GEOS-Chem model simulation, the surface



PM$_{2.5}$ observation and meteorological reanalysis dataset. Finally, we examine the ubiquity of this
phenomenon in plain regions in China and propose a conceptual model, providing detailed vertical
insights into the enhancement of nocturnal surface PM$_{2.5}$.

## 95 2 Data and methods

### 96 2.1 Multi-lidar system

A multi-lidar system was installed on the mobile observation vehicle. The vehicle, a modified 7-seater
Mercedes-Benz sport utility vehicle, was equipped with three lidar instruments mounted on steel bars at
the rear for stability. The mobile observation routes were primarily on flat highways, and the speed was
controlled to remain around 80 km/h to minimize the impact of frequent changes in speed and vehicle
bumps on the measurement results.
The multi-lidar system (Everise Technology Ltd., Beijing) consisted of a 3D visual scanning micro pulse
lidar (EV-Lidar-CAM), a twirling Raman temperature profile lidar (TRL20), a Doppler wind profile lidar
(WINDVIEW10), a global positioning system (GPS). The 3D visual scanning micro pulse lidar had a
detection range of up to 30 km, a temporal resolution of 1 minute, and a vertical resolution of 15 m. The
3D lidar emitted a 532 nm laser beam vertically, which is scattered by aerosol particles in the atmosphere.
The backscattered signal is utilized to calculate the aerosol extinction coefficient and depolarization ratio
profile. The extinction coefficient increases with higher particle pollution concentrations, while the
depolarization ratio can distinguish between spherical and non-spherical particles based on their size and
shape. The Doppler wind profile lidar provides a temporal resolution of 1 minute and a vertical resolution
of 50 m. It emits a rotating 1545 nm laser beam and measures the Doppler shift produced by the laser's
backscattered signal as it passes through airborne particles such as dust, water droplets in clouds and fog,
polluted aerosols, salt crystals, and biomass-burning aerosols to derive the horizontal and vertical wind
speeds at any height. The Raman temperature profile lidar, based on Raman scattering theory, calculates
atmospheric temperature by detecting the rotational Raman scattering signal of nitrogen or oxygen
molecules in the atmosphere. Operating at a 532 nm wavelength, it has a temporal resolution of 5 minutes
and a vertical resolution of 60 m. The quality of the data obtained by the lidar system was checked by
the Integrated Environmental Meteorological Observation Vehicle before deployment. The results
showed a percentage difference of less than 15% between the lidar system data and the data provided by
the Shenzhen Meteorological Tower, demonstrating the high accuracy of the lidar instrument (Xu et al.,



2022). Previous studies had utilized this lidar system and demonstrated its reliability (Xu et al., 2018; He
et al., 2021). The mobile observation vehicle and multi-lidar system are shown in Figure 1(a).

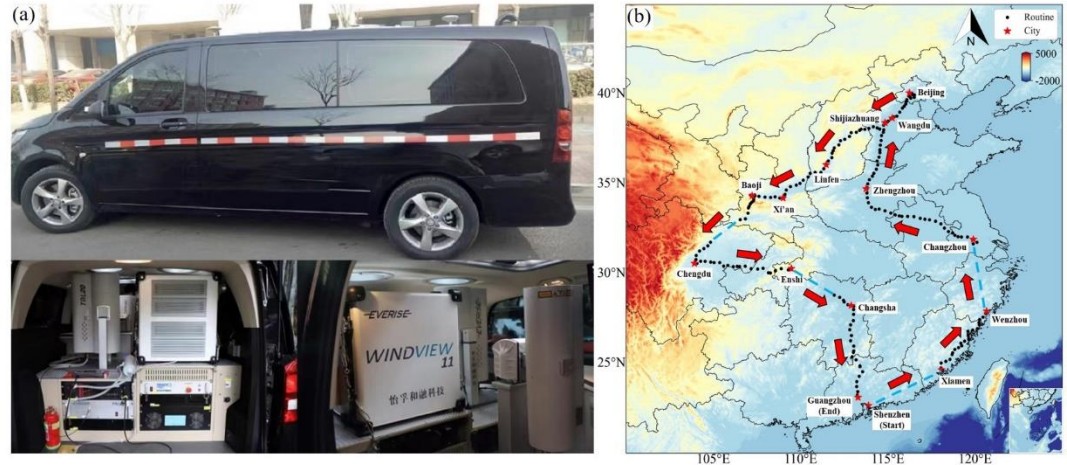

**Figure 1.** (a) The mobile observation vehicle and multi-lidar system. (b) The mobile observation route and stopover
cities, the blue dotted line shows the sections of missing data.
**2.2 The route of nationwide mobile observation**
To investigate the distribution characteristics of particulate matter during winter in different regions in
China, the Integrated Environmental Meteorological Observation Vehicle of Sun Yat-sen University was
utilized to conduct the first nationwide mobile observation campaign. The campaign, which lasted 43
days and covered approximately 11,000 km, started in Shenzhen on 30 November, 2018 and ended in
Guangzhou on 11 January, 2019. This campaign surveyed the $PM_{2.5}$ vertical profiles across 15 cities,
including Shenzhen, Xiamen, Wenzhou, Changzhou, Zhengzhou, Wangdu, Beijing, Shijiazhuang,
Linfen, Xi'an, Baoji, Chengdu, Enshi, Changsha and Guangzhou. The observation route and stopover
cities are shown in Figure 1(b). Due to the precipitation, there were no observations between Shenzhen-
Xiamen and Wenzhou-Changzhou, while some GPS data were missing between Beijing-Chengdu and
Enshi-Changsha.
To compare the vertical distribution characteristics of particulate matter in different regions, we
conducted fixed-point observations for several pollution days in four representative cities in the East
China region (Changzhou), North China Plain (Wangdu), Guanzhong Basin (Xi'an), and Sichuan Basin
(Chengdu). The dates and duration of the fixed-point observations are presented in Table 1. In the



following analysis, only the data obtained in the four fixed-point measurements are used since it has an
enough time duration to show the vertical variation of $PM_{2.5}$.

**Table 1.** Date and cities of fixed-point observations

| Date | Cities | Coordinate | Landform |
|------|--------|-----------|----------|
| 2018.12.11-2018.12.14 | Changzhou | 119.97°E, 31.83°N | Plain area |
| 2018.12.18-2018.12.22 | Wangdu | 115.25°E, 38.67°N | Plain area |
| 2018.12.31-2019.01.02 | Xi'an | 109.01°E, 34.22°N | Basin area |
| 2019.01.04-2019.01.09 | Chengdu | 103.92°E, 30.58°N | Basin area |


### 2.3 Surface $PM_{2.5}$ data and ERA5 reanalysis data

The nationwide hourly observations of surface $PM_{2.5}$ in China are obtained from the China National
Environmental Monitoring Center (CNEMC) network (https://quotsoft.net/air, last accessed: March 2nd,
2023). Here, we used the hourly $PM_{2.5}$ concentration data from the whole winter of 2018 (Dec. 2018 –
Feb. 2019) and selected data from the closest monitoring station to show the change in surface $PM_{2.5}$
concentration at the four observation sites.
The spatial distribution of daily average surface $PM_{2.5}$ concentration is obtained from the TAP team
(http://tapdata.org.cn), with spatial resolution of 10 km. Based on machine learning algorithms and multi-
source data information, the TAP team has built a multi-source data fusion system that integrates ground
observation data, satellite remote sensing information, high-resolution emission inventories, air quality
model simulations and other multi-source information (Geng et al., 2021a; Xiao et al., 2021a). In addition
to the observation data, we also apply the three-dimensional meteorological data from ERA5 dataset for
the winter of 2018 (https://quotsoft.net/air, last accessed: March 2nd, 2023) (Munoz-Sabater et al., 2021).
This dataset contained temperature, horizontal and vertical wind speed, and direction at pressure levels,
as well as two-dimensional data including sea-level pressure and 2-m temperature. The ERA5 dataset is
the fifth generation of the European Centre for Medium-Range Weather Forecasts (ECMWF)
atmospheric reanalysis of the global climate. The ERA5 dataset has a horizontal resolution of
0.25°×0.25°, a vertical resolution of 25 hPa, and a temporal resolution of 1 h.



**2.4 HYSPLIT backward trajectory model**

The Hybrid Single Particle Lagrangian Integrated Trajectory Model (HYSPLIT) (Stein et al., 2015), developed by NOAA Air Resources Laboratory, which is a valuable tool for simulating the movement of air mass and the transport of pollutants in the atmosphere, is used in our study to obtain the sources of particulate matter at different heights. Altitudes of 100, 500, and 1000 m were set as the end points of the trajectories, the meteorological input for the trajectory model was the FNL dataset, and each trajectory was calculated for 24 h duration.

**2.5 GEOS-Chem model description**

Given the short-term (less than one week) fixed-point observation duration of the mobile observation vehicle in each city, we employ the global three-dimensional chemical transport model GEOS-Chem version 13.3.1 to interpret the vertical observations (available at https://github.com/geoschem/GCClassic/tree/13.3.1, last assessed: March 2nd, 2023, (Bey et al., 2001)) and to simulate the distribution of particulate matter concentrations during winter 2018 in China. We perform the nested-grid version of GEOS-Chem simulation at a spatial of 0.5° (latitude) × 0.625° (longitude) resolution over East Asia (60-150°E, 11°S-55°N), The model has 47 vertical layers with 18 layers in the below 3 km. Boundary chemical conditions for the nested models are archived from a consistent global simulation run at 4° latitude × 5° longitude resolution. Meteorological input is from the Modern-Era Retrospective analysis for Research and Application version 2 (MERRA-2) (Gelaro et al., 2017). We conduct the model simulation from 2018/11-2019/02 with the first one month as spin-up. The model mechanisms and emissions mostly follow our previous study (Wang et al., 2022). In short, the GEOS-Chem model describes a comprehensive stratospheric and tropospheric ozone–$NO_x$–VOCs– aerosol–halogen chemical mechanism (Wang et al., 1998; Park et al., 2004; Parrella et al., 2012; Mao et al., 2013). Photolysis rates are computed using the Fast-JX scheme (Bian and Prather. 2002). Advection of tracers in GEOS-Chem is accomplished through TPCORE advection algorithm. Boundary layer mixing process is described in (Lin and McElroy. 2010). Dry and wet deposition of both gas and aerosols is considered (Wesely. 1989; Zhang et al., 2001). We apply the latest version of the Community Emissions Data System (CEDSv2) anthropogenic emissions inventory (O'Rourke et al., 2021), in which the emissions over China have been adjusted to align with the Multi-resolution Emission Inventory for China (MEIC) inventory (Zheng et al., 2018).



## 3 Results and discussions

### 3.1 The observation of nocturnal PM$_{2.5}$ enhancement in plain areas

During the fixed-point observation in Changzhou, we observed a typical surface PM$_{2.5}$ concentration enhancement event starting at 4:00 and lasting until 10:00 on 13 December. As shown in Figure 2(a), the concentration of PM$_{2.5}$ increased from 69 to 151 μg/m$^3$. Figure 2(b-c) showed the spatiotemporal distribution of the extinction coefficient and depolarization ratio. There was a clear layer with low extinction coefficient below 500 m from 16:00 on 12 December to 4:00 on 13 December, indicating low PM$_{2.5}$ concentration near the surface. Meanwhile, an aerosol layer with high extinction coefficient of about 0.7 km$^{-1}$ appeared at 500-1,000 m. Figure 2(d-e) depicted the west winds prevailed the layer of 500-1,000 m with a wind speed (WS) of about 7 m/s. Based on the daily averaged concentration of PM$_{2.5}$ on 12 December shown in Figure S1, the west area of the observation site in Changzhou suffered from severe air pollution with the concentration of PM$_{2.5}$ exceeding 150 μg/m$^3$. Under the strong forcing of the west winds, the regional transport of aerosol from the west of Changzhou was detected, leading to a high extinction coefficient layer at 500-1,000 m. The spatiotemporal distribution of the vertical velocity in Figure 2(e) indicated the dominant updraft winds in the ABL, which was conducive to the suspension of pollutants at 500-1,000 m.

However, the prevailing winds at 500-1,000 m shifted to the northwest/north after 4:00 on 13 December. By 8:00, the north wind dominated in the ABL. The change in wind direction affected the transport process of pollutants at 500-1,000 m, after which the transport basically disappeared. Meanwhile, the downdraft winds dominated above 500 m (Figure 2(e)) and the aerosol layer suspended at the 500-1,000 m began to gradually transport and diffuse downward into the lower layer of ABL, which enhanced the nocturnal surface PM$_{2.5}$ concentration. Noteworthy, after 4:00 on 13 December, the surface temperature was close to the temperature at 950 hPa, suggesting that the structure of the ABL was stable and was conducive to the accumulation of the PM$_{2.5}$.

The sea surface field showed the cold high-pressure system moved southeast with increasing strength from 20:00, 12 December to 8:00, 13 December (Figure S2). The change in the synoptic weather system was accompanied by a cold frontal passage. The cold frontal passage was inferred to start at about 4:00, 13 December and last about 4 hours, which was further illustrated by the clockwise rotation of the horizontal wind from ground to upper layer (Shi et al., 2022) and the transition from updrafts to downdrafts, the observation site was located behind the cold front after 4:00 where the descending



movements dominated. Under the influence of the subsidence, the pollutants transported by the west
advection diffused downward to the low layer and further aggravated the local air quality.

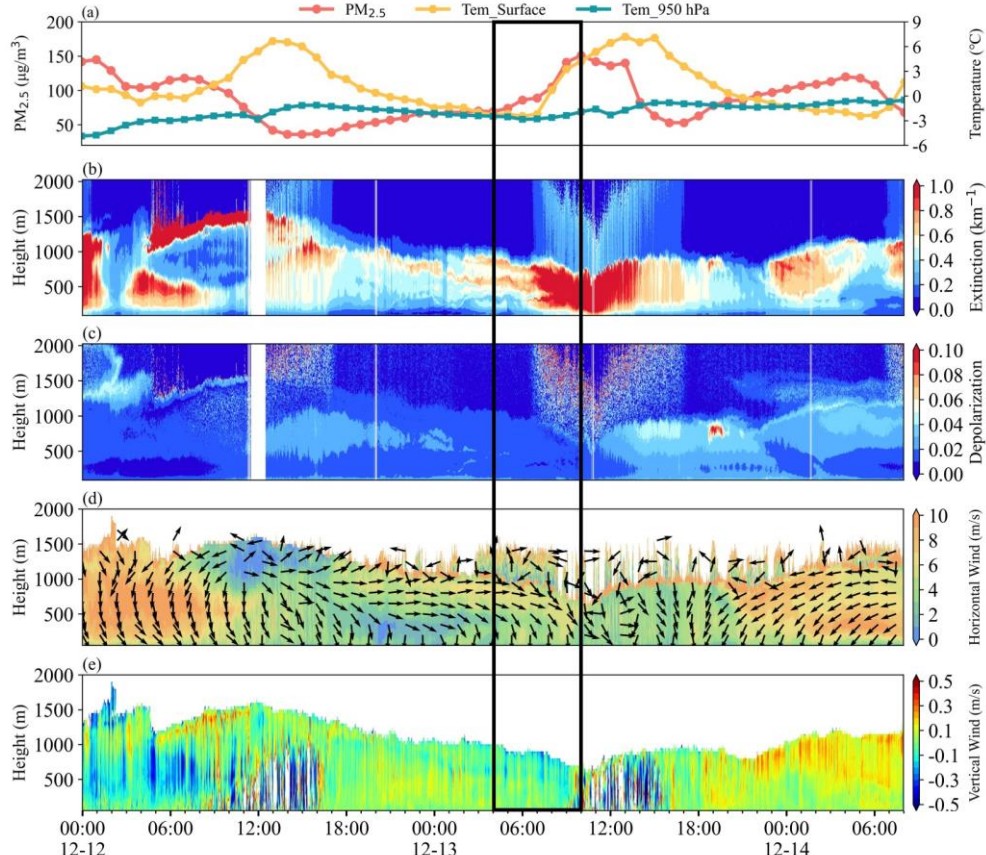


**Figure 2.** (a) Surface PM$_{2.5}$ concentration, surface temperature and 950 hPa temperature, (b) Extinction coefficient,
(c) Depolarization ratio, (d) Horizontal wind, and (e) Vertical wind, during the observation in Changzhou from 12
December to 14 December. The black box indicated the nocturnal surface PM$_{2.5}$ enhanced event.
After 8:00, the concentration of surface PM$_{2.5}$ increased rapidly and peaked at around 10:00, the
extinction coefficient below 1,000 m also reached a high level with 1.0 km$^{-1}$ at the same time and the
depolarization remained about 0.01. The surface temperature began to rise and the convective ABL
developed rapidly, which enhanced the vertical mixing and resulted in the rapid increase in surface PM$_{2.5}$
(Zhou et al., 2023). And the north winds following the passage of the cold front dominated in the ABL
after 8:00, which could bring pollution from the NCP to the YRD (Kang et al., 2019; Huang et al., 2020).





Therefore, we attribute the increase in the concentration of surface $PM_{2.5}$ from 4:00 to 10:00 to the
combination of the subsidence behind the cold front before 8:00, vertical mixing caused by the
development of the convective ABL, and the transport by the north winds.
We also found similar nocturnal surface $PM_{2.5}$ enhancement events during the fixed-point observation in
Wangdu, on 19 December and 21 December respectively (Figure 3(a)). The concentration of $PM_{2.5}$
started to enhance at 1:00, 19 December, and meanwhile, the layer of pollutants above 1,000 m started
to transport and diffuse to the lower layer of ABL which was reflected by the change of the extinction
coefficient shown in Figure 3(b). Unfortunately, due to the instrument malfunction, the wind profile data
was unavailable and we used the ERA5 data instead,  which previously showed good consistent with the
observation of with Doppler wind lidar (Shi et al., 2022). As shown in Figure 3(d), from 10:00, 18
December to 0:00, 19 December, southwest winds prevailed above 1,000 m and the WS exceeded 8 m/s,
a persistent southerly wind could result in severe air pollution in the NCP (Cai et al., 2017; Callahan et
al., 2019; Zhang et al., 2019a). The wind forced the regional advection of pollutants from the south region
suffered from serious air pollution (Figure S3) to the observation site. Meanwhile, the updrafts dominated
in the ABL which facilitated the suspension of pollutants in the upper layer. After 0:00, 19 December,
as the cold high-pressure system moved southwest accompanied by a cold front (Figure S4), the
prevailing winds above 1,000 m shifted to northwest gradually and downdrafts dominated behind the
cold frontal passage. The changes in the horizontal and vertical wind fields caused the advection of
pollutants to disappear basically and the pollutants layer suspended above 1,000 m began to transport
and diffuse downward to the low layer of ABL. The passage of the cold front at 0:00, 19 December,
lasted for 4 hours, and the subsidence behind the cold front caused the pollutants to diffuse downward,
enhancing the concentration of nocturnal $PM_{2.5}$.

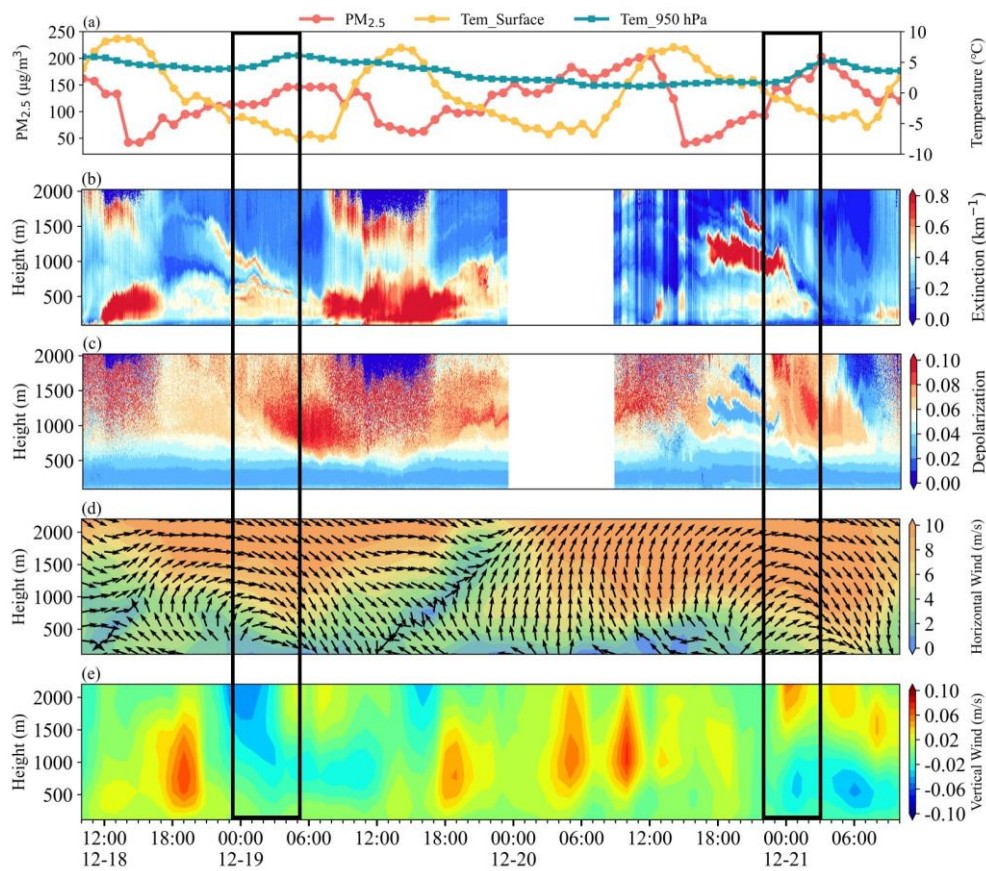

**Figure 3.** (a) Surface PM$_{2.5}$ concentration, surface temperature and 950 hPa temperature, (b) Extinction coefficient, (c) Depolarization ratio, (d) Horizontal wind, and (e) Vertical wind, during the observation in Wangdu from 18 December to 21 December. The black boxes indicated the Nocturnal PM$_{2.5}$ enhancement events.

The pattern of the nocturnal surface PM$_{2.5}$ enhancement event on 21 December was highly similar to that on 19 December. However, the pollutant advection process lasted a longer duration which started at 16:00, 20 December and ended at 0:00, 21 December (Figure 3(b)), and the WS of the southwest wind above 1,000 m exceeded 12 m/s meeting the standard of the low-level jet (LLJ) (Kraus et al., 1985; Hu et al., 2013). The area south of the observation site in Wangdu suffered from more severe air pollution with the concentration of PM$_{2.5}$ exceeding 300 μg/m$^3$ (Figure S5). Under the strong forcing of the southwester LLJ and the updrafts depicted in Figure 3(d-e), an aerosol layer with high extinction coefficient exceeding 2 km$^{-1}$ formed and was suspended at 1,000-1,500 m from 16:00, 20 December to



0:00, 21 December. Meanwhile, Figure 3(c) showed that the layer with low depolarization was consistent
with the layer with high extinction coefficient, further confirmed the role of transportation.
After 0:00, the wind direction of LLJ began to change due to the southeasterly movement of the high-
pressure system accompanied a cold front (Figure S6). The passage of cold front started at 0:00, 21
December and lasted for 4 hours, after which the downdrafts dominated below 1,500 m (Figure 3(e)),
and the northwester LLJ no longer transported pollutants from the southern area but greatly enhanced
the turbulent mixing (Shi et al., 2022). Under the influence of the turbulence generated by LLJ and
subsidence behind the cold front, the aerosol-rich layer suspended at 1,000-1,500 m was gradually
transported and diffused downward into the lower layer of ABL, ultimately enhancing the concentration
of surface $PM_{2.5}$, which was consistent with the result reported by (Shi et al., 2022), with the secondary
inorganic aerosol increasing simultaneously during the subsidence process as observed by the tethered
balloon.
Noteworthy, when both nocturnal surface $PM_{2.5}$ enhancement events in Wangdu occurred, the
temperature at 950 hPa showed an increasing trend as a result of the heating of the air by compression as
it descended, while the surface temperature continuously declined (Figure 3(a)). The opposite variation
of surface temperature and temperature at 950 hPa stabilized the lower atmosphere. The stronger
inversion layer was probably induced by subsidence (Carlson and Stull. 1986). With the more stably
atmospheric layer and inversion during subsidence, the concentration of surface $PM_{2.5}$ enhanced
(Gramsch et al., 2014; Largeron and Staquet. 2016).
**3.2 Transport-Nocturnal $PM_{2.5}$ Enhancement by Subsidence events**
During the fixed-point observation, we found the causes of three nocturnal $PM_{2.5}$ enhancement events in
different cities were similar. The processes included three steps: First, the horizontal winds with high
wind speed forced the transport of pollutants from the upstream region, while the updrafts dominated,
both resulting in the formation and suspension of an aerosol layer with high extinction coefficient at the
high layer of the ABL. Then, under the influence of the southeasterly movement of high-pressure system
and the passage of the cold front, the horizontal wind direction shifted to north or northwest and the
downdrafts became dominant. Finally, the transport of pollutants disappeared due to the change of wind
direction, and under the subsidence behind the cold front, the aerosol-rich layer suspended at high layer
was gradually transported and diffused downward into the lower layer of the ABL, ultimately enhancing



the concentration of nocturnal PM$_{2.5}$. Here, we defined this pollution pattern as T-NPES (Transport-
Nocturnal PM$_{2.5}$ Enhancement by Subsidence) events.
To investigate the occurrence frequency of T-NPES events, we employed the GEOS-Chem model to
simulate the distribution of particulate matter concentrations in China during the whole winter of 2018
(Dec. 2018 – Feb. 2019). We utilized the simulated PM$_{2.5}$ at 950 hPa and 900 hPa to represent the high-
altitude PM$_{2.5}$ concentration. We selected the closest grid data of the wind field data, 950 hPa and 2-m
temperature data from ERA5 dataset to the observation site in Changzhou and Wangdu to show the
meteorological condition. By analysing the hourly concentration variation of PM$_{2.5}$ and the distribution
of the wind field during the three months of winter 2018 in Changzhou and Wangdu, we found 11 typical
T-NPES events in Changzhou accounting for 12.2% and 18 T-NPES events in Wangdu accounting for
18%, which indicated that the T-NPES events were a relatively common phenomenon in the two cities.
Figure 4 showed the average pattern of all T-NPES events in Changzhou, the trend of the simulated PM$_{2.5}$
was consistent with the observation, confirming the credibility of the simulations. As shown in Figure
4(a), the enhancement of nocturnal surface PM$_{2.5}$ started at 21:00, when there was no significant
enhancement in anthropogenic PM$_{2.5}$ emissions, while the high-altitude PM$_{2.5}$ represented by PM$_{2.5}$ at
900 hPa and 950 hPa started to decrease, which was consistent with the observed event in Changzhou
described in Section 3.1. According to the distribution of wind field (Figure 4(b)), west winds with high
wind speed prevailed the layer above 1,000 m from 0:00 to about 18:00, which was conducive to the
transport of pollutants. And the updrafts dominated from 0:00 to 12:00, forcing the pollutants suspending
in the upper layer, which was reflected by the enhancing PM$_{2.5}$ concentration at high-altitude (Figure
4(a)). Despite the downdrafts dominated after 12:00, there was no immediate reduction in PM$_{2.5}$
concentration at high-altitude, which might be related to the fact that the horizontal wind direction had
not changed, and the transport of pollutants continued. A brief updraft before 21:00 suspended the
pollutants at high-altitude. After 21:00, northwester winds and downdrafts dominated in the ABL and
the high-altitude PM$_{2.5}$ began to gradually transport and diffuse downward causing the enhancement of
surface concentration of PM$_{2.5}$, and this process continued until 4:00 in the next day. The surface
temperature and the temperature at 950 hPa gradually approached, which is consistent with the observed
case in Changzhou, indicating that the structure of the ABL was stable and was conducive to the
accumulation of the PM$_{2.5}$. As shown in Figure S7, the average sea level pressure indicated that the
southeasterly movement of the high-pressure system and the passage of cold front, which resulted in the
shift in wind direction and subsidence behind the cold front, were the main causes of the T-NPES events.



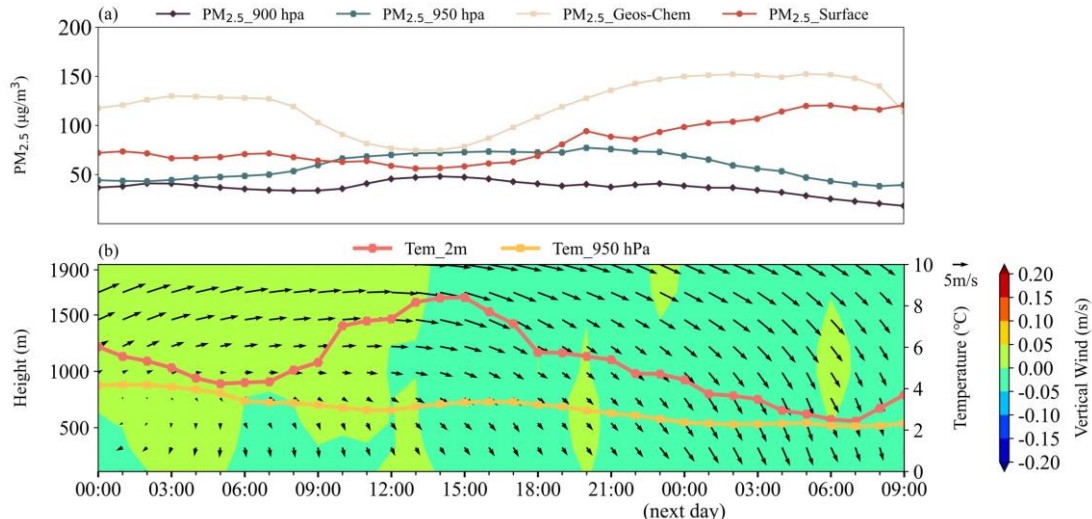

**Figure 4.** The average for all T-NPES events in Changzhou. (a) The concentration of PM$_{2.5}$ at different levels, surface PM$_{2.5}$ of observation (red line), surface PM$_{2.5}$ of simulation (blue line), PM$_{2.5}$ at 900 hPa and 950 hPa. (b) The horizontal winds (arrows), the vertical winds (shaded), temperature at 2 m and temperature at 950 hPa.

Figure 5 showed the average pattern of all T-NPES events in Wangdu, which was similar to that in Changzhou. Figure 5(a) demonstrated that the trend of simulated PM$_{2.5}$ was consistent with the observation before 22:00 but was different thereafter. The trend of high-altitude PM$_{2.5}$ was increasing before 15:00 due to the transport of pollutants by prevailing southwester horizontal winds and the dominant of updrafts which suspended the aerosol shown in the Figure 5(b). After 18:00, the prevailing winds began to turn to northwest and ultimately turn to north at 0:00 in the next day, while a brief updraft between 18:00 and 20:00 suspended the pollutants at high-altitude. The ABL was dominated by the northwester winds and downdrafts after 21:00. Simultaneously, the high-altitude PM$_{2.5}$ began to gradually transport and diffuse downward causing the enhancement of surface concentration of PM$_{2.5}$. The temperature at 950 hPa increased and the surface temperature declined (Figure 5(b)), which agreed with the two observation examples in Wangdu. The opposite variation of temperature at different height stabilized the ABL and further enhanced the concentration of PM$_{2.5}$. By analysing the weather circulation patterns, the causes of the T-NPES events were the same with those in Changzhou and were attributed to the southeasterly movement of high-pressure system and the passage of the cold front (Figure S8). Overall, the average patterns of T-NPES events in Changzhou and Wangdu were essentially in good agreement with the three cases of T-NPES in the two cities. But there were still slight differences, such





as the change of Wangdu caused by the movement of high-pressure lasted a longer time in the average
situation and the start time of subsidence behind the cold front was also not consistent, which were due
to each T-NPES event was not exactly the same.

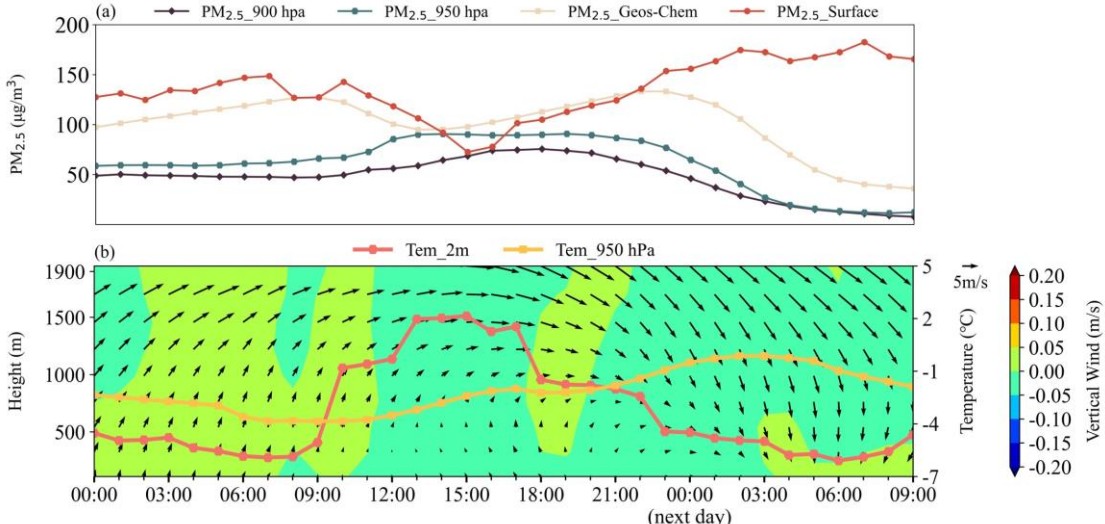

**Figure 5.** The average for all T-NPES events in Wangdu. (a) The concentration of PM$_{2.5}$ at different levels, surface
PM$_{2.5}$ of observation (red line), surface PM$_{2.5}$ of simulation (blue line), PM$_{2.5}$ at 900 hPa and 950 hPa. (b) The
horizontal winds (arrows), the vertical winds (shaded), temperature at 2 m and temperature at 950 hPa
**3.3 The universality of T-NPES events in eastern China**
Despite the mobile observation vehicle had no observations in other cities of the NCP, the YRD and the
Loess Plateau, we could still utilize the simulated data and the ERA5 data to investigate the universality
of T-NPES events occurrence in other cities. We selected Shijiazhuang, Beijing and Tianjin as
represented cities of the NCP, Shanghai and Nanjing as represented cities of the YRD and Taiyuan,
Linfen as represented cities of the Loess Plateau. We found the similar pattern of T-NPES events in all
these cities. However, these T-NPES events in different cities had some differences in detail. Here we
divided the T-NPES events into four types based on the status of PM$_{2.5}$ after T-NPES events. More
information on the types, frequency of the T-NPES events and their percentage of the winter 2018 was
shown in Table 2.
The typical representation of Type 1 was shown in Figure S9, the characteristic of Type 1 was that the
southwester winds transported the pollutants in high-altitude of the ABL, then the wind direction shifted
to north and downdrafts dominated, finally, pollutants in high-altitude diffused into lower layer causing





the surface PM$_{2.5}$ enhanced. However, after T-NPES event, the north wind near the ground was not strong
enough to remove the pollutants, causing the high level of PM$_{2.5}$ lasting the next day morning and may
resulting in aggravation of the air pollution in the following day.  The characteristic of T-NPES event of
Type 2 was basically consistent with Type 1. However, after the T-NPES event, as north winds became
stronger, pollutants were rapidly removed, resulting in a clean boundary layer throughout (Figure S10).
Even when the pollutants were removed more quickly by stronger north winds, the subsidence process
might not be observed. Type 1 and Type 2 were both observed in the NCP cities, while Type 1
predominated in Wangdu and Shijiazhuang, and Type 2 in Beijing and Tianjin.
**Table 2.** Statistics of the T-NPES events in cities during Dec. 2018 – Feb. 2019

| Area | Type | City | Frequency (days) | Percentages |
|---|---|---|---|---|
| NCP | Type 1 and 2 | Wangdu | 18 | 20.0% |
| | | Shijiazhuang | 18 | 20.0% |
| | | Beijing | 13 | 14.4% |
| | | Tianjin | 14 | 15.6% |
| YRD | Type 3 | Changzhou | 11 | 12.2% |
| | | Shanghai | 7 | 7.8% |
| | | Nanjing | 8 | 8.9% |
| Loess Plateau | Type 4 | Linfen | 18 | 20.0% |
| | | Taiyuan | 13 | 14.4% |


Figure S11 showed the typical representation of Type 3. The prevailing wind transporting pollutants was
not southwest but west and the start and end of the T-NPES event were later than for Type 1 and 2. After
the T-NPES event, the increase of 2-m temperature and the development of convective ABL led to the
vertical mixing and the increase of surface PM$_{2.5}$. Additionally, the stronger north wind might transport
the pollutants from the NCP to the YRD. The Type 3 was similar to the example in Changzhou in Section
3.1 and indicative of a typical pattern in the YRD cities.
The typical representation of Type 4, which was mainly occurred in the Loess Plateau cities, was shown
in Figure S12. During the T-NPES event, the change of wind direction was only observed above 1,500
m while the wind speed below was so weak that the shirt in wind direction was not significant, which
was significantly different from the wind field of other three types. The reason for the difference between

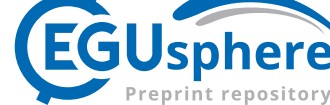

Type 4 and other types was mainly related to the topography of the Loess Plateau, which has a blocking
effect on the movement of high-pressure system. Noteworthy, after the analysis of these T-NPES events
in different cities in China, we suggested that the T-NPES events were a common pattern of the nocturnal
$PM_{2.5}$ enhancement, but did not always have an impact on the air pollution of the following day. The
pollution levels on the following day depended more on the strength of the cold front, local pollution
conditions, the structure of ABL and regional transportation. Further quantification is needed to
determine the relationship between the T-NPES events and the pollution levels on the following day.
Based on these mentioned above, we suggested that the T-NPES events were a common phenomenon in
winter in plain areas such as the NCP and the YRD. A conceptual model was thus developed and shown
in Figure 6, there were the transportation of aerosol by the horizontal winds above 1,000 m and the
updrafts dominated before night, which was conducive to the formation and suspension of the aerosol
layer. Then, as the southeasterly movement of the high-pressure system and the passage of the cold front
at about the time of midnight, the wind direction began to turn to north/northwest, causing the aerosol
diluted. Finally, the downdrafts dominated in the ABL and the LLJ might enhance the turbulent. Under
the influence of subsidence behind the cold front and turbulence, the depth of the aerosol layer suspended
above 1,000 m began to decrease and the pollutants gradually transported and diffused downwards into
the lower layer of the ABL, enhancing the concentration of surface $PM_{2.5}$.

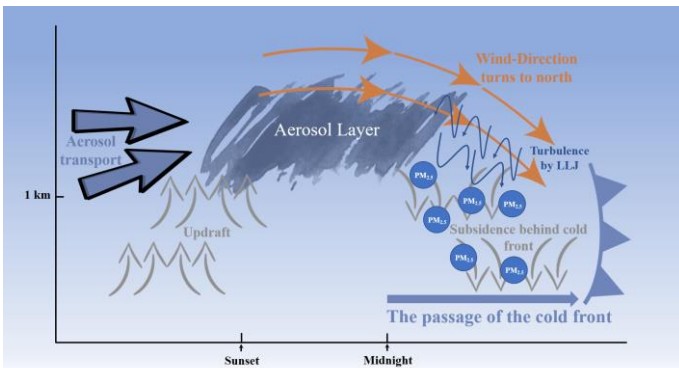


**Figure 6.** Conceptual scheme of the T-NPES events
**3.4 No T-NPES event occurred in Basin areas**
We further checked the fix-point measurement in Xi'an and Chengdu, two cities with typical basin
topography. The results indicated that there were essentially no T-NPES events in either city, suggesting



the conceptual did not work. Figure 7(a) indicated that the concentration of surface $PM_{2.5}$ had no
enhancement during the night from 23:00 on 31 December to 4:00 on 1 January, and from 22:00 on 1
January to 3:00 on 2 January in Xi'an. $PM_{2.5}$ remained at a high concentration, while the extinction
coefficient did not show the subsidence process, suggesting that the T-NPES events were not common
here.
Taking the night of 31 December as an example, from 18:00 on 31 December to 4:00 on 1 January, the
concentration of surface $PM_{2.5}$ increased before 23:00 and then stabilized at high values, while the
extinction coefficient remained a high level with about 1.0-1.2 $km^{-1}$ near 500 m. As shown in Figure 7(d),
from 18:00, 31 December to 6:00, 1 January, a light wind layer appeared below 1,000 m, with ~1 m/s.
Such a static and stable condition was conducive to the accumulation of locally generated particulate
matter near the ground, causing the concentration of $PM_{2.5}$ to enhance between 18:00 and 23:00 on 31
December and the formation and maintenance of the aerosol layer at about 500 m. Noteworthy, the wind
direction at low layer was southeaster, while it was the opposite northwester at about 1,000 m, which
was the typical characteristic of mountain-valley breeze circulation. The dominance of downdrafts below
500 m suggested that Xi'an was in the upper area of the nocturnal mountain-valley breeze circulation.
The mountain-valley breeze circulation could only be observed when the background WS was relatively
weak, which further indicated a stable structure of the ABL. The example on 1 January was similar to
the above one, with the extinction coefficient reaching 2 $km^{-1}$ and depolarization ratio decreasing after
21:00 due to the hygroscopic growth of aerosol by the rise in relative humidity.

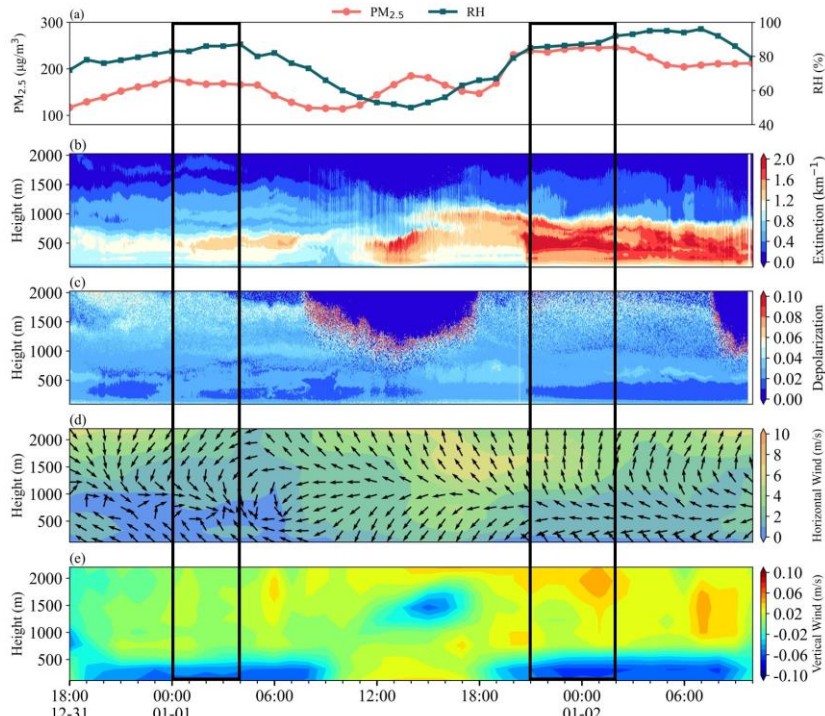


**Figure 7.** (a) Surface PM₂.₅ concentration and relative humidity, (b) Extinction coefficient, (c) Depolarization ratio,
(d) Horizontal wind, and (e) Vertical wind, during the observation in Xi'an from 18:00, 31 December to 10:00, 2
January. The two black boxes were the time period to be studied.

Due to the topography of the basin in Xi'an, the mountain-valley breeze circulation, or the horizontal
winds with lower WS always dominated in the ABL, which was not conducive to the transport and
dispersion of particulate matter. The stable structure of the ABL resulted in the particulate matter
accumulated in the low layer, which was the main feature of the nocturnal particulate matter distribution
in Xi'an.
Figure 8 showed that the concentration of surface PM₂.₅ also had no significant enhancement but
remained a high value over nighttime in Chengdu. The distribution of the extinction coefficient in the
two black boxes presented double-layer structure, one layer near 250 m and another layer suspended at
about 500 m. Meanwhile, the wind field exhibited typical mountain-valley breeze circulation, as shown
in the two black boxes in Figure 8(d), which presented westerly wind near 250 m and southeasterly wind
above 500 m. The variation wind direction due to the mountain-valley breeze circulation at different
layer might be responsible for the double-layer of particulate matter. Figure S13 illustrated the backward



trajectories when the double-layer appeared. The layer of particulate matter at about 100 m might have
originated from the southwest area of Chengdu, whereas the layer of particulate matter at 500 m and
1,000 m might have originated from the northeast area of Chengdu. The different sources of particulate
matter were consistent with the mountain-valley breeze circulation in Chengdu, further demonstrating
the dominance of mountain-valley breeze in the static and stable ABL at night.
The orange box in Figure 8 indicated that the distribution of particulate matter in the ABL of Chengdu
under the dominance of northeasterly winds with high WS. Both extinction coefficient and the
depolarization ratio showed a stratified structure, with the extinction coefficient initially higher below
750 m and lower above 750 m, whereas the depolarization ratio exhibited the opposite trend. The main
cause of this phenomenon was that the different sources of particulate matter in the two layers. Under
the influence of the dominant updrafts, local emissions with high depolarization ratio were transported
upwards, while the lower layer was occupied by particulate matter with lower depolarization ratio
transported by the northeasterly wind. As the continuous transport of the northeasterly wind, the entire
ABL was occupied by transported particulate matter with a high extinction coefficient and a low
depolarization ratio.
Due to the short time during the fixed-point observation period, it is difficult to make a universal
conclusion that no T-NEPS occurs in basin regions. Therefore, we further checked the surface and model
simulation data of the two basin cities for three months in winter 2018. We found that, unlike the plain
area, the T-NEPS events were almost never observed in the basin regions. It confirmed that the
conceptual model was indeed not applicable in the basin area. This was mainly attributed to the fact that
the movement of the weather system was blocked by the mountains surrounding the basin. Therefore,
the movement of the high-pressure system and the passage of the cold front had a weak impact on the
basin region. Without the downdrafts and the shift in wind direction associated with the movement of
the high-pressure system and the passage of the cold front, the structure of the ABL between Xi'an and
Chengdu was relatively stable, making it difficult for particulate matter to be transported and diffused,
and thus accumulate in the ABL at night. During the three months, we found that the wind field in Xi'an
was dominated by the light winds, while in Chengdu there were two states: one is dominated by light
winds and the other by strong northeasterly wind. Fortunately, our fixed-point observations had captured
these typical processes indeed. In addition, considering the wind fields in basin cities were mainly
dominated by light winds, which was the main characteristic in basin area (Bei et al., 2016; Shu et al.,
2021) and was similar to the wind fields below 1,500 m in Taiyuan and Linfen of the Type 4. Therefore,



we suggested that the Loess Plateau cities might serve as a crucial transitional zone between the plains
and the basin as introduced in Section 3.3. In summary, the conceptual model of T-NPES events was
applicable to the plain areas which were more influenced by the movement of weather system in winter,
such as the NCP and YRD, but not to the basin areas.

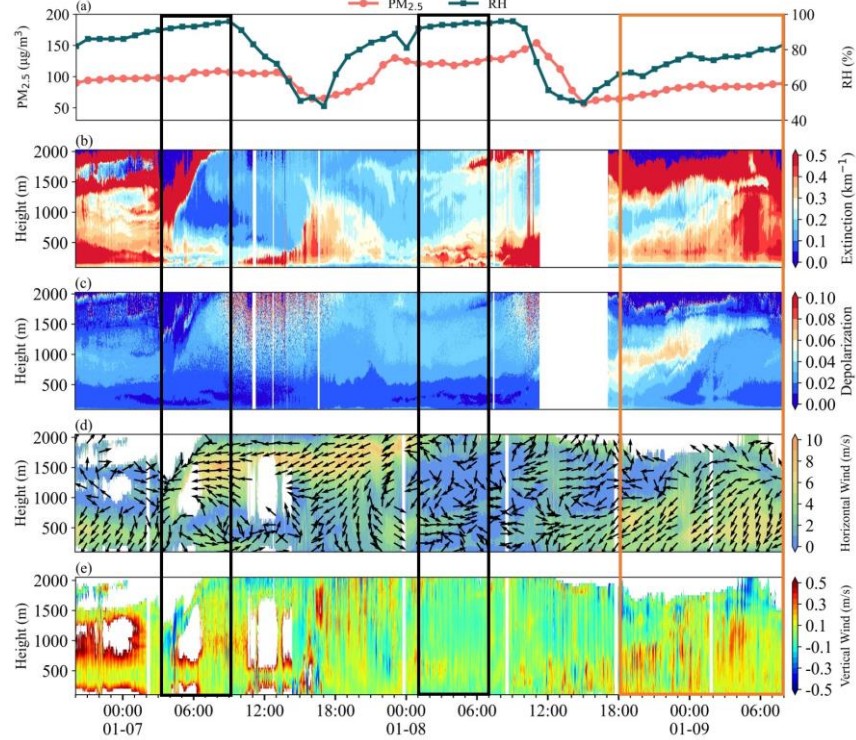


**Figure 8.** (a) Surface PM$_{2.5}$ concentration and relative humidity, (b) Extinction coefficient, (c) Depolarization ratio,
(d) Horizontal wind, and (e) Vertical wind, during the observation in Chengdu from 20:00, 7 January to 8:00, 10
January. The two black boxes were the time period of the double-layer structure, the orange box was the time period
to be studied.
**4 Conclusions and outlook**
In this study, we reveal that the T-NPES is a relatively common and important pathway that causes PM$_{2.5}$
pollution in the surface layer in the plain areas in winter China. The fixed-point observations in
Changzhou and Wangdu demonstrated that the T-NPES was associated with the subsidence of particulate
matter in the upper layer due to the movement of high-pressure and the passage of the cold front. Model



simulations further confirmed the ubiquity of T-NPES events in plain areas, despite these event types
varied case by case. However, the observations in Xi'an and Chengdu indicated that the event was less
occurred in the basin areas, as the impact of weather system was weakened by the obstruction of
mountains surrounding the basin. In further studies, more multi-lidar measurement should be conducted
in other cities in the plains and basin areas to look insight to the detailed mechanism of T-NPES events.
In addition, more works are urgently needed to uncover the vertical profiles of chemical components of
the particulate matter, since it may also be affected by the coupling of physical and chemical processes.
**Code/Data availability.** The datasets used in this study are available at:
https://doi.org/10.5281/zenodo.8368944 (Wang et al., 2023).
**Author contributions.** H.C.W. and S.J.F designed the study. Y.M.W. and H.C.W. analysed the data,
H.L.W. and X.L. provided the GEOS-Chem model simulation results, Y.M.W. and H.C.W. wrote the
paper with input from all coauthors.
**Competing interests**. The authors declare that they have no conflicts of interest.
**Acknowledgments**. The authors gratefully acknowledge the NOAA Air Resources Laboratory (ARL)
for the provision of the HYSPLIT transport and dispersion model used in this study.
**Financial support**. This research has been supported by the Guangdong Major Project of Basic and
Applied Basic Research (grant no. 2020B0301030004), the Guangdong science and technology plan
project (grant no. 2019B121201002), and the National Natural Science Foundation of China (grant no.

508 42175111).

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
