# Peer review of "Measurement report: Nocturnal subsidence behind the cold"

_EGUsphere, 2023_

## Author Comment (AC1)

**Reviewer #1**

**Comment [1-1]:** General comments: This paper installs a multi-lidar system that can measure aerosol extinction coefficient, wind direction, wind speed, and temperature on a vehicle and measures meteorological phenomena that can affect changes in PM2.5 concentration in real time in major cities in China. This is an analysis paper. Through analysis of observation results, the researchers revealed that T-NPES (Transport-Nocturnal PM2.5 Enhancement by Subsidence) is a relatively common and important pathway causing PM2.5 pollution in the surface layer of China's winter plains. It is believed that the above research results were possible thanks to the multi-lidar system. It is considered to be academically meaningful to reveal that T-NPES is the main cause of high concentration PM2.5. However, the main content of the paper is a simple structure that repeats the analysis of observation results, and no new results were found other than T-NPES. This is disappointing considering the long observation period and observation area.

Although this paper is judged to have low academic value to be published in this journal, it is ultimately judged that it can be published in this journal because it is a research result that can only be observed using a multi-lidar system.

**Response [1-1]: We thank Prof. Noh for the comments and suggestions. All of them have been implemented in the revised manuscript, such as adding a more detailed description of the measurement system, and more discussion about the nocturnal PM$_{2.5}$ enhancement in Plain regions in China. Please see our itemized blue responses below for more details.**

**Comment [1-2]:**

1. It is deemed necessary to add essential information about the multi-lidar system.

2. ex) In 3D visual scanning micro pulse lidar, information on measurement wavelength and lowest observable altitude must be added

**Response [1-2]: We have added the essential information about the multi-lidar system in Table S1 (attached below). And we improved the description of the multi-lidar system as follows:**
**Line 107: "The 3D lidar used an Nd: YAG laser to emit a 532 nm laser beam at a repetition frequency of 2500 Hz, which is scattered by aerosol particles in the atmosphere."**

**Line 112: The Doppler wind profile lidar: "It emits a rotating 1545 nm laser beam using a 10 kHz repetition rate fiber-pulse laser and measures the Doppler shift produced by the laser's backscattered signal as it passes through airborne particles such as dust, water droplets in clouds and fog, polluted aerosols, salt crystals, and biomass-burning aerosols to derive the horizontal and vertical wind speeds at any height."**

**Line 118: The Raman temperature profile lidar: "Operating at a 532 nm wavelength by an Nd: YAG laser at a repetition frequency of 20 Hz, it has a temporal resolution of 5 minutes and a vertical resolution of 60 m."**

**Table S1.** Detailed parameters for the multi-lidar system

| Lidar | Variable | Wavelength | Spatial and time resolution | Lowest observable altitude |
|---|---|---|---|---|
| 3D visual scanning micro pulse lidar | Extinction coefficient, depolarization ratio | 532 nm | 15 m/1 min | 30 m |
| Doppler wind profile lidar | Wind speed and direction profiles | 1545 nm | 50 m/1 min | 40 m |
| Raman temperature profile lidar | Temperature profiles | 532 nm | 60 m/5 min | 60 m |

**Comment [1-3]:** Figure 6 is difficult to read, so resolution, etc. needs to be improved.

**Response [1-3]: Thanks for your suggestion. We have made the following adjustments to improve the comprehensibility of Figure 6 and attached below:**

1. **We add the illustration to show the plain area at the bottom of Figure 6.**
2. **We change the color of the background and the arrows to make the picture more vivid.**
3. **We remove the "1 km" and use the "Altitude" to make the vertical axis clearer.**
4. **We use the "Particulates" instead of "Aerosol" more accurately describe the subject of the study, and improve the legend of the transport of "Particulates" and the "Particulates layer".**
5. **We add the description of the key T-NPES process as "PM₂.₅ enhancement by subsidence behind the cold front"**
6. **We use a gradual change legend to represent the passage of the cold front better.**
7. **We add "3 steps" to better summarize the steps of the T-NPES event and improve the comprehensibility of the conceptual scheme.**

[Figure]

**Figure 6.** Conceptual scheme of the T-NPES events

---

## Author Comment (AC2)

**Reviewer #2**

**Comment [2-1]:** General comments: This study reveal that the T-NPES is a relatively common and important pathway that causes PM2.5 pollution in the surface layer in the plain areas in winter China. Comprehensive mobile-lidar data and surface monitoring data are presented and analyzed to support the conclusion. The mechanism and potential regions of this phenomenon taking place is well demonstrated. Overall, the data presented herein is robust and the mechanism of T-NPES induced nocturnal PM2.5 increase is well explained. It is well suited to *ACP* journal and suggested to be published after addressing my following concerns.

**Response [2-1]: We thank the reviewer for the positive and valuable comments. All of them have been implemented in the revised manuscript. Please see our itemized blue responses below.**

**Comment [2-2]:** Line 43: A comma should be placed before 'such as' instead of a period.
**Response [2-2]: Corrected accordingly**

**Comment [2-3]:** Figure 1(b): The label of black dots is suggested to be named as 'route'.
**Response [2-3]: Thanks. We have revised the name of the label of black dots as 'route'. The new Figure 1 is shown below:**

[Figure]

**Comment [2-4]:** Section 2.1: In addition to the description of instruments used in this study, in terms of clarification, the method of data processing and QA/QC procedures should be at least briefly presented herein. And I suggest listing all parameters measured by lidar system and other surface station in a table with resolution, uncertainty and other related features for better readability.
**Response [2-4]: Thank you for the suggestion. Quality control of lidar data is always difficult. We have introduced the method of QA/QC procedures in the Section 2.1 as "The quality of the data obtained by the lidar system was checked by the Integrated Environmental Meteorological Observation Vehicle before deployment. The results showed a percentage difference of less than 15% between the lidar system data and the data provided by the Shenzhen Meteorological Tower, demonstrating the high accuracy of the lidar instrument (Xu et al., 2022)." And here we supplied the data processing as the following: "Data during the instrument malfunction, below the blind zone and in rainy weather had been excluded.". In addition, we have followed your suggestion and**

**supplied more parameters about the lidar system to the Table S1.**

**Table S1.** Detailed parameters for the multi-lidar system

| Lidar | Variable | Wavelength | Spatial and time resolution | Lowest observable altitude |
|---|---|---|---|---|
| 3D visual scanning micro pulse lidar | Extinction coefficient, depolarization ratio | 532 nm | 15 m/1 min | 30 m |
| Doppler wind profile lidar | Wind speed and direction profiles | 1545 nm | 50 m/1 min | 40 m |
| Raman temperature profile lidar | Temperature profiles | 532 nm | 60 m/5 min | 60 m |

**Comment [2-5]:** Line 332~334: What is the reason of the discrepancy between observed PM2.5 and simulated PM2.5 by geoschem? Can the model reproduce the same wind field and temperature as observed one which, as stated above in the manuscript, cause the subsidence of air mass aloft?

**Response [2-5]: The input meteorological data in the GEOS-Chem model is from the Modern-Era Retrospective analysis for Research and Application version 2 (MERRA-2), so the model results are generally similar to the observations and are able to reproduce well wind field and temperature variations similar to those observed by the lidar system. We supplied the comparison of model results with observations in Figure S1 and the following description in Section 2.5: "Figure S1 showed the comparison of model results with observations for monthly mean PM$_{2.5}$, and the correlation coefficients between model and observation were about 0.6, which meant that the model results provided a relatively good reproduction of the observations."**

**The reason for the discrepancy between observed and simulated PM$_{2.5}$ by GEOS-Chem could be due to the relatively low resolution of the model, resulting in a large difference in some of the individual T-NPES cases, which has an impact on the average results shown in the Figure 5.**

[Figure]

**Figure S1.** The comparison of model results (color map) with observations (color points) for monthly mean PM$_{2.5}$ values and the correlation coefficient (R) between model and observation. (a) 2018.12, (b) 2019.01, (c) 2019.02

**Comment [2-6]:** Line 386: Please change 'shirt' to 'shift'.
**Response [2-6]: Corrected accordingly.**

**Comment [2-7]:** Line 384~404: It seems like the type 4 T-NPES does not follow the same illustrated pattern as the other three and it is also different from the conceptual plot depicted in Fig 6. Maybe more information related to type 4 should be added into Fig 6.
**Response [2-7]: Yes, as we have mentioned in the Section 3.4: "We suggested that the Loess Plateau cities might serve as a crucial transitional zone between the plains and the basin as introduced in Section 3.3." Type 4 is a transition type of T-NPES between the plains and the basins, and the Loess Plateau is at a critical point in terms of the occurrence of T-NPES events, so Type 4 is not identical to the other three types. However, shifts in wind direction and transport of high-altitude PM$_{2.5}$ to the surface, as well as the enhancement of surface PM$_{2.5}$, as described in the conceptual scheme are still observed in Type 4. Therefore, we still classify it as a typical type of T-NPES.**
**We improve the conceptual scheme (Figure 6) as below to better represent the T-NPES events in the plain area, where the T-NPES are more characterized.**

[Figure]

**Figure 6.** Conceptual scheme of the T-NPES events

**Comment [2-8]:** Although the authors presented several case studies to elaborately explain the pattern of T-NPES and its contribution on the increase of nocturnal surface PM2.5 concentrations, as stated in the text, the percentage of occurrence of this phenomenon was less than 20%. In that way, it comes to me that there should be some cases with increasing nocturnal PM2.5 in the surface during non-T-NPES condition, or with typical T-NPES event not causing increasing nocturnal PM2.5 in the surface. I suggest making some comparison among these three different cases, which might help to better illustrate the features and significance of T-NPES on PM2.5 pollution.
**Response [2-8]: Thanks for your suggestion. This is an aspect we had not well considered. Here we further take Wangdu and Changzhou as the representative cities to conduct a comparison among these three different cases, we highlight that the T-NPES make a large contribution to the nocturnal PM2.5 enhancement and partly responsible for the pollution event, while we should note that the**

T-NPES event not always cause pollution, we also show that the event can also clean away the aerosol by a fast-north wind, this may be explained by the fast wind shifts and the north wind speed is high to remove aerosol quickly before the increase in surface PM$_{2.5}$ is observed. Here we added the discussion in the Section 3.3 as following:

Line 409: "To look insights into the mechanism of nocturnal PM$_{2.5}$ enhancement, we systematically documented instances of nocturnal PM$_{2.5}$ enhancement during the winter of 2018 in Wangdu and Changzhou according to the surface PM$_{2.5}$ observation. We identified 48 such events in Wangdu and 27 in Changzhou, with proportions of T-NPES events of 37.5% and 40.7%, respectively. The results implied that T-NPES represents merely one among multiple pathways contributing to the nocturnal PM$_{2.5}$ enhancement. We checked the nocturnal PM$_{2.5}$ enhancement events that not caused by T-NPES in Wangdu, the dominant wind field distributions within the ABL were southerly or characterized by static light wind, which indicated that the nocturnal PM$_{2.5}$ enhancement might result from either horizontal transport from polluted regions in the southern areas or the local accumulation of particulates in the stable ABL. In the nocturnal PM$_{2.5}$ enhancement events of non-T-NPES condition in Changzhou, higher wind speeds in the ABL and predominantly from the northern and southwestern, which indicated the nocturnal PM$_{2.5}$ enhancement might result from horizontal transport from the NCP (Huang et al., 2020) or caused by other reasons. For example, from the perspective of chemical formation, the nocturnal atmospheric oxidation may elevate the nighttime aerosol concentration (Wang et al., 2023; Yan et al., 2023). In addition, we found the T-NPES event not always cause nocturnal PM$_{2.5}$ increase, in a few cases, the strong north wind following the cold front play a role in remove the aerosol. In summary, the T-NPES just represents one vertical transport mechanism that can collectively contributes to the enhancement of nocturnal PM$_{2.5}$ with other physical and chemical processes (Zhao et al., 2023). Further understanding of the coupling effect of transportation as well as the chemical formation to the nocturnal PM$_{2.5}$ enhancement is thus highly needed."

**References:**

Huang, X., Ding, A.J., Wang, Z.L., Ding, K., Gao, J., Chai, F.H., Fu, C.B.: Amplified transboundary transport of haze by aerosol-boundary layer interaction in China. Nat. Geosci., 13, 428-+. http://doi.org/10.1038/s41561-020-0583-4, 2020

Wang, H.C., Wang, H.L., Lu, X., Lu, K.D., Zhang, L., Tham, Y.J., Shi, Z.B., Aikin, K., Fan, S.J., Brown, S.S., Zhang, Y.H.: Increased night-time oxidation over China despite widespread decrease across the globe. Nat. Geosci., 16, 217-+. http://doi.org/10.1038/s41561-022-01122-x, 2023

Xu, X.Q., Xie, J.L., Li, Y.M., Miao, S.J., Fan, S.J.: Measurement report: Vehicle-based multi-lidar observational study of the effect of meteorological elements on the three-dimensional distribution of particles in the western Guangdong-Hong Kong-Macao Greater Bay Area. Atmos. Chem. Phys., 22, 139-153. http://doi.org/10.5194/acp-22-139-2022, 2022

Yan, C., Tham, Y.J., Nie, W., Xia, M., Wang, H.C., Guo, Y.S., Ma, W., Zhan, J.L., Hua, C.J., Li, Y.Y., Deng, C.J., Li, Y.R., Zheng, F.X., Chen, X., Li, Q.Y., Zhang, G., Mahajan, A.S., Cuevas, C.A., Huang, D.D., Wang, Z., Sun, Y.L., Saiz-Lopez, A., Bianchi, F., Kerminen, V.M., Worsnop, D.R., Donahue, N.M., Jiang, J.K., Liu, Y.C., Ding, A.J., Kulmala, M.: Increasing contribution of nighttime nitrogen chemistry to wintertime haze formation in Beijing observed during COVID-19 lockdowns. Nat. Geosci., 16, 975-+. http://doi.org/10.1038/s41561-023-01285-1, 2023

Zhao, X.X., Zhao, X.J., Liu, P.F., Chen, D., Zhang, C.L., Xue, C.Y., Liu, J.F., Xu, J., Mu, Y.J.: Transport Pathways of Nitrate Formed from Nocturnal $N_2O_5$ Hydrolysis Aloft to the Ground Level in Winter North China Plain. Environ. Sci. Technol. http://doi.org/10.1021/acs.est.3c00086, 2023